Do patterns of insect mortality in temperate and tropical zones have broader implications for insect ecology and pest management?

Pinto José R. L. 1
http://orcid.org/0000-0003-3489-4754 Fernandes Odair A. 1
http://orcid.org/0000-0003-2949-7375 Higley Leon G. 2
Peterson Robert K. D. 3 bpeterson@montana.edu
1 Department of Agricultural Production Sciences, São Paulo State University , Jaboticabal, São Paulo , Brazil
2 School of Natural Resources, University of Nebraska-Lincoln , Lincoln, Nebraska , United States
3 Land Resources and Environmental Sciences, Montana State University , Bozeman, Montana , United States
Andrew Nigel
Electronic publication date: 2022 Apr 25
Publication date: 2022
Volume: 10
Electronic Location ID: e13340
Received 2021 Dec 9; Accepted 2022 Apr 5
Copyright: © 2022 Pinto et al.
Copyright year: 2022
Copyright holder: Pinto et al.
License: This is an open access article distributed under the terms of the Creative Commons Attribution License, which permits unrestricted use, distribution, reproduction and adaptation in any medium and for any purpose provided that it is properly attributed. For attribution, the original author(s), title, publication source (PeerJ) and either DOI or URL of the article must be cited.
License URL: https://creativecommons.org/licenses/by/4.0/

Keywords: Insect demography, Biogeography, Population dynamics, Multiple decrement life table, Biological control, Parasitoid, Predator

Funding: Montana State University Montana Agricultural Experiment Station This research was funded in part by Montana State University and the Montana Agricultural Experiment Station. The funders had no role in study design, data collection and analysis, decision to publish, or preparation of the manuscript.

==============================
Background

Understanding how biotic and abiotic factors affect insect mortality is crucial for both fundamental knowledge of population ecology and for successful pest management. However, because these factors are difficult to quantify and interpret, patterns and dynamics of insect mortality remain unclear, especially comparative mortality across climate zones. Life table analysis provides robust information for quantifying population mortality and population parameters.

Methods

In this study, we estimated cause-of-death probabilities and irreplaceable mortality (the portion of mortality that cannot be replaced by another cause or combination of causes) using a Multiple Decrement Life Table (MDLT) analysis of 268 insect life tables from 107 peer-reviewed journal articles. In particular, we analyzed insect mortality between temperate and tropical climate zones.

Results

Surprisingly, our results suggest that non-natural enemy factors (abiotic) were the major source of insect mortality in both temperate and tropical zones. In addition, we observed that irreplaceable mortality from predators in tropical zones was 3.7-fold greater than in temperate zones. In contrast, irreplaceable mortality from parasitoids and pathogens was low and not different between temperate and tropical zones. Surprisingly, we did not observe differences in natural enemy and non-natural enemy factors based on whether the insect species was native or non-native. We suggest that characterizing predation should be a high priority in tropical conditions. Furthermore, because mortality from parasitoids was low in both tropical and temperate zones, this mortality needs to be better understood, especially as it relates to biological control and integrated pest management.

Introduction

Insect ecology is the study of the numerous interactions that occur between insects and the biotic and abiotic components that occur simultaneously in the environment (Price et al., 2011). It follows, then, that insect population dynamics are influenced by biotic and abiotic factors (Krebs, 2014). Consequently, from an applied ecological standpoint, understanding how these factors affect insect mortality is crucial for successful insect pest management.

However, simultaneous action of biotic and abiotic mortality factors is difficult to characterize, quantify, and interpret. Therefore, robust methodological techniques are required for the study of population ecology of insects. Life tables provide valuable information for describing and quantifying population mortality and population parameters that are crucial to understanding dynamics of insect populations in numerous ecological systems (Carey, 1989; Cornell & Hawkins, 1995; Royama, 2001; Carey & Roach, 2020). This is because biological demography offers experimental models, conceptual tools, and evolutionary perspectives using well-studied and validated comparative techniques to answer questions about the demographic characteristics of species (Carey, 2001; Carey & Vaupel, 2019; Carey & Roach, 2020).

To better estimate the effect of a specific cause of mortality, we have to consider its interaction with competing causes of mortality (Weiss, 1990; Peterson et al., 2009; Clouston et al., 2016; Carey & Roach, 2020). Drawing on decades of use by human demographers, Carey (1989) proposed the use of the Multiple Decrement Life Table (MDLT) for ecological questions. As part of the MDLT, Carey (1989) also proposed using the relatively well-understood concept of irreplaceable mortality, which is the mortality rate from a cause of death that cannot be replaced by another cause of death or combination of multiple causes of death operating at the same time. As such, values for percentage irreplaceable mortality represent actualized reductions in populations. With this information, it is possible to estimate the contribution of each cause of mortality and better understand mortality dynamics in ecosystems.

Several ecological hypotheses suggest that biotic and abiotic factors could act differently between climate zones (Dyer & Coley, 2002; Schemske et al., 2009). In addition, several researchers have applied life table approaches to understand patterns of insect mortality in different climate zones (Cornell & Hawkins, 1995; Hawkins, Cornell & Hochberg, 1997; Dyer & Coley, 2002; Peterson et al., 2009; Kareithi et al., 2019). Generally, most of these analyses focused on a specific group of insects and did not use actual sampling measurements of mortality from different causes or presented few studies in tropical or subtropical zones. Therefore, patterns of insect mortality between climate zones remain unclear.

Using a MDLT analysis from 73 previously published insect life tables, Peterson et al. (2009) found that the irreplaceable mortality from non-natural enemy factors (mostly abiotic factors) was significantly greater than all types of natural enemy mortality factors. Furthermore, they observed low irreplaceable mortality from parasitoids and pathogens. However, they only used life tables that included three or more mortality factors and only a few studies from tropical zones. Therefore, they concluded that a necessary next step was to evaluate the importance of environmental stability on mortality. Consequently, in this paper, we expand the analysis to evaluate insect mortality in a much broader context, and by doing so we can compare insect mortality dynamics and patterns between temperate and tropical zones.

Materials and Methods

Data acquisition

We created MDLTs with data from 107 peer-reviewed journal articles. The publications represented studies from 1954 to 2021 and include data from 24 countries across all continents except Antarctica (Fig. 1). In total, 51 papers presented data from temperate zones and 56 presented data from tropical zones (Table S1). Temperate and tropical zones are defined following Marsh & Kaufman (2012). Using that classification scheme, we designated temperate sites as those between 35° and 90° latitude and tropical sites as those between 0° and 34°. We chose not to create a “subtropical” group because it is highly problematic to define the subtropics. The number of possible climatic or ecological definitions of the subtropics is unlimited and there is no mechanism to ensure universal agreement.

Figure 1 Distribution of life table studies used for data collection.

The data from the 107 publications include 268 field-based life tables that were analyzed (Table S2). To identify appropriate publications, life tables, and data that met our criteria, we used the bibliography search engine Web of Science (Clarivate Analytics, Inc., London, United Kingdom) and Google Scholar (Google, Inc., Menlo Park, CA, USA).

The data we analyzed consisted of 85 phytophagous and one non-phytophagous insect species from 58 different plant hosts distributed in three major categories: crop (55.17%), forest (34.48%), and non-crop (10.34%). The insect species were distributed in the following orders: Coleoptera 21.50%, Diptera 4.67%, Hemiptera 13.08%, Hymenoptera 6.54%, and Lepidoptera 54.21%. From the 107 publications, 66 studies were related to native species and 41 studies were related to non-native species, defined according to CABI (Invasive Species Compendium, www.cabi.org).

The field-based studies we selected for analysis were those that met the basic criteria for analysis of MDLTs (Peterson et al., 2009; Davis, Peterson & Higley, 2011; Carey & Roach, 2020). All life tables used in our study included multiple age-specific mortality factors (e.g., predation, parasitism, disease, desiccation, physiological disorders, egg infertility) which allowed assignment in MDLTs and application of cause-of-death analysis by each factor. With the aid of the spreadsheet-based program M-DEC (Davis, Peterson & Higley, 2011), we calculated the percentage mortality for each mortality cause in the presence of other mortality causes, and subsequently, the irreplaceable mortality. For this, variables were defined as: x the life stage index, lx the number of individuals alive at each x, kx the number at the beginning of each x, dx the total number of deaths in each stage, alx the fraction of the cohort living at the beginning of the stage (starting at 1.0 for the first stage and calculated by alx−1 − adx−1), adix fraction of deaths attributable to one cause, adx fractions of all deaths from all causes (ad1x + ad2x + …. + ad5x), and, aqx stage-specific probability of death within that stage calculated by the sum of the probability of dying from all listed causes (dx/kx). The irreplaceable mortality for each specific cause was estimated using an elimination-of-cause method following Carey (1993), Peterson et al. (2009), Davis, Peterson & Higley (2011), Buteler et al. (2015), Carey & Roach (2020).

The probability for cause of death in the absence of other causes was estimated using a quadratic solution (Carey, 1993) within the M-DEC program (Davis, Peterson & Higley, 2011). Elimination-of-cause analysis relies on the probability of surviving each source of mortality (Px) and its complement (1 − qx) where (1 − q1) x. . .x (1 − qn) is the chance of jointly surviving a set of mortality factors and its complement, 1 − [(1 − q1) x. . .x (1 − qn)], is the chance of jointly dying from a set of mortality factors. To estimate mortality in the absence of one or more factors, two simultaneous equations with two unknowns are used. For example, by expressing q1 in terms of q2, D1, and D2 (the fraction of all individuals observed that died of cause 1 and 2), this yields the quadratic equation aq22 + bq2 + c = 0, where a = D1, b = −(D1 + D2) and c = D2(D1 + D2). The value of q2 can be found by substituting a, b, and c into the quadratic formula.

We grouped the causes of mortality into main categories: predators, parasitoids, pathogens, and non-natural enemy factors (i.e., rainfall, wind, desiccation, failure to establish). The category of “non-natural enemy factors” is admittedly an awkwardly termed designation because it does not exclusively consist of abiotic factors, but also could include host-plant factors. We also created other groupings of interest for comparisons such as all natural enemies (parasitoids, predators, and pathogens) and insect natural enemies (i.e., natural enemies of insects that are insects). As also done by Peterson et al. (2009), to avoid bias we standardized the number of life tables per study because some studies presented multiple life tables, whereas others presented only one table. We used a “per unique life table” statistic for comparisons. This means that when a study presented more than one life tables that only varied over site or year or both, we calculated the mean value of these tables so that they could be suitable for comparison with a study that only presented one life table (Table S2). However, in addition to the selection criteria adopted by Peterson et al. (2009) in which they only used life tables that included three or more mortality factors, we included life tables that only had two reported mortality factors (n = 58) (e.g., parasitism and unknown). This is because our intent was to evaluate more thoroughly the mortality dynamics associated with parasitoids. Therefore, we created and expanded the database to enable evaluation of insect mortality in a broader context.

Data analysis

We conducted analysis of variance with the percentage mortality in the presence of other mortality causes and the irreplaceable mortality data “per unique life table” generated from temperate (n = 51) and tropical zones (n = 56) using mixed models (α = 0.05) (proc MIXED, SAS Institute Inc, 2015). Also, mixed models were used to verify the relationship between the percentage mortality in the presence of other mortality causes and the irreplaceable mortality in the created group “all natural enemies” and “all non-natural enemy factors” when the insect species was native or non-native from the study site. We chose not to use regression models based on latitude because if we used them with Bartlett’s test and arrived at the conclusion that the variances are heterogeneous, we would have faced the problem of how to best analyze the data and estimate effects (treatment means and/or treatment differences) correctly accounting for the heterogeneous variances (different data adjustments and transformations for each analyzed group). Proc MIXED obviates these problems; the mixed model equations are defined quite generally with an Error (Residual) Variance-Covariance matrix R.

For multiple comparisons among mortality data collected from all zones (n = 107), Tukey’s test (α = 0.05) was applied (proc MIXED, SAS Institute Inc, 2015). To statistically classify the irreplaceable mortality observed between the study site locations (n = 24) in clusters, the non-hierarchical multivariate k-means clusters analysis was conducted (STATISTICA version 7.0 Statsoft Inc, 2004). This method was chosen because it minimizes the variance of the data within each cluster and seeks to group the locations according to the similarity between them, represented here by the six key variables of interest: all natural enemies (1), insect natural enemies (2), parasitoids (3), all non-natural enemy factors (4), all predators (5), all pathogens (6). Irreplaceable mortality data were standardized through the z-score formula (z = (x − μ)/σ), where x is the raw score, μ is the population mean, and σ is the population standard deviation. The similarity between the clusters was measured by the Euclidean distance between each study site locations and the centroid of the cluster. When forming clusters of study sites with similar irreplaceable mortality to assist in visualizing patterns, k = 4 best represented a classificatory structure of groups. Therefore, considering the number of clusters equal to four, the k-means clustering method was applied to ordering countries where the studies were conducted such that the similarity and the differences within clusters were maximum. As a measure of similarity between countries, the Euclidean distance was used in a six-dimensional space, which corresponds to the irreplaceable mortalities of the six factors studied. For the cluster number validation, k number was defined using the ‘‘elbow’’ method, in which we examined the plot of fusion coefficients against the number of clusters (Webb & Copsey, 2011). Then, we compared and described these clusters based on their performance.

It is important to note that the presented data come from the evaluation of 268 life tables representing a variety of insect species, locations, data collection methods, and research questions. Therefore, the results should be cautiously interpreted.

Results

The mean irreplaceable mortality (±SE) by all non-natural enemy factors of 30.96 ± 2.56% was significantly greater than for all types of natural-enemy mortality factors (21.52 ± 2.21%) (F5,510 = 23.93, P < 0.0001) (Fig. 2). In contrast, the lowest irreplaceable mortality values were 5.03 ± 1.61% for pathogens, 6.69 ± 1.03% for parasitoids, and 13.77 ± 2.37% for predators. We also found similar irreplaceable mortality between all natural enemies (20.52 ± 2.21%) and insect natural enemies (17.15 ± 2.12%) (F5,510 = 1.10, P = 0.2730). Similar results were observed for mortality in the presence of other factors, in which all non-natural enemy factors (52.38 ± 2.72%) was the major source of mortality. In contrast, parasitoids (13.35 ± 1.57%) and pathogens (8.57 ± 2.09%) had the lowest percentages of mortality (Fig. 2).

Figure 2 Irreplaceable mortality and mortality in the presence of other factors by mortality categories.

Columns followed by the same letter in each mortality category are not different (α = 0.05); values inside each column represent means.

Our results show that the irreplaceable mortality caused by all natural enemies, all parasitoids, pathogens, and all non-natural enemy factors in temperate zones was not statistically different than that evaluated in tropical zones (Table 1, Fig. 3). Conversely, life-table data for tropical zones suggested higher irreplaceable mortality from all insect natural enemies (F1,103 = 4.64, P = 0.0333) and predators (F1,59 = 13.48, P = 0.0005) than temperate zones (Table 1, Fig. 3). Furthermore, even with a reduced number of analyzed life tables for temperate (n = 116) compared to tropical (n = 152), the mean overall mortality, typically egg to adult, from all causes, was similar between temperate and tropical zones. When expressing mean mortality in the presence of other factors between tropical and temperate zones, we observed statistical evidence of differences for all natural enemies (F1,102 = 5.97, P = 0.0163), all insect natural enemies (F1,101 = 5.83, P = 0.0175), and predators (F1,61 = 25.73, P < 0.0001).

Figure 3 Irreplaceable mortality and mortality in the presence of other factors by mortality categories in temperate and tropical zones (and asterisk (*) indicates significant differences (α = 0.05) for each mortality category; values inside each column represent means).

Table 1 Mean irreplaceable mortality and mortality in the presence of other factors by mortality categories in temperate and tropical zones.

Irreplaceable mortality	
Mortality categories	Temperate zone	Tropical zone	
All non-natural enemy	31.93 ± 3.87 a	30.07 ± 3.42 a	
All natural enemies	17.16 ± 2.56 a	23.51 ± 3.47 a	
All insect natural enemies	12.12 ± 2.08 b	20.79 ± 3.43 a	
All parasitoids	5.04 ± 1.04 a	5.76 ± 2.55 a	
Predators	5.71 ± 1.38 b	21.08 ± 3.95 a	
Pathogens	2.02 ± 0.63 a	3.75 ± 1.18 a	
All-cause mortality	84.50 ± 2.70 a	84.11 ± 2.69 a	
Mortality in the presence of other factors	
Mortality categories	Temperate zone	Tropical zone	
All non-natural enemy	56.78 ± 4.04 a	48.30 ± 3.63 a	
All natural enemies	26.34 ± 3.00 b	37.71 ± 3.55 a	
All insect natural enemies	22.57 ± 2.95 b	34.03 ± 3.70 a	
All parasitoids	9.02 ± 1.38 a	11.10 ± 1.37 a	
Predators	12.68 ± 2.55 b	38.69 ± 4.60 a	
Pathogens	11.35 ± 4.61 a	5.53 ± 1.36 a	
All-cause mortality	81.48 ± 3.25 a	84.11 ± 2.69 a	
Note:

Means (±SE) followed by the same letter within rows for each species were not statistically diﬀerent by the F test (p > 0.05).

No significant difference was observed between the irreplaceable mortality for all natural enemies for native (20.26 ± 2.99%) and non-native insect species (19.84 ± 3.11%) (F1,105 = 0.01, P = 0.9222). Similar results were observed for irreplaceable mortality from all non-natural enemy factors (F1,104 = 0.18, P = 0.6682) for native species (31.94 ± 3.42%) and non-native species (29.73 ± 3.83%). Therefore, there was no statistical evidence for differences in irreplaceable mortality between native and non-native species.

Our results suggest that, except for parasitoids, all other irreplaceable mortalities were important for group ordering (α = 0.05) (Table 2). Cluster 1 (Fig. 4) consisted of only tropical countries (Brazil, Malaysia, Mexico, and Nigeria). This cluster is classified for high irreplaceable mortality for all natural enemies, all insect natural enemies, and all predators, but low irreplaceable mortality for the other variables. Cluster 2 consisted of more countries (China, England, Eritrea, Ethiopia, France, Honduras, India, Japan, New Zealand, Philippines, Samoa, and South Africa), which shares low values for all irreplaceable mortality factors. Cluster 3 ordered countries classified by extremes in temperature (Australia, Israel, Malawi, Spain, Switzerland, and United States), characterized by high irreplaceable mortality for non-natural enemy factors. Finally, Cluster 4 represented countries that displayed high irreplaceable mortality for pathogens (Canada and Kenya).

Figure 4 Standardized means of irreplaceable mortality within each variable (mortality groups) in each group according to analysis of non-hierarchical clusters k-means.

Table 2 Analysis variance for each variable of the groups formed by the non-hierarchical analysis of k-means clusters.

Variable	Sum of squares between groups	Degrees of freedom	Sum of squares within groups	Degrees of freedom	Fc	Prob.	
All natural enemies	18.17	3	4.83	20	25.07	<0.0001	
All insect natural enemies	18.64	3	4.36	20	28.49	<0.0001	
All parasitoids	2.02	3	20.98	20	0.64	0.5975	
All non-natural enemy factors	16.44	3	6.56	20	16.72	<0.0001	
All predators	15.37	3	7.63	20	13.43	0.0005	
All pathogens	20.90	3	2.09	20	66.38	<0.0001	
Note:

Fc: calculated F value. Prob.: probability of obtaining a value of F ≥ Fc.

Discussion

One of our aims for this study was to characterize the dynamics of different insect mortality factors across climate zones. Our results surprisingly suggest that non-natural enemy factors (i.e., abiotic and host-plant factors) are the major source of insect mortality in both temperate and tropical zones. Somewhat unsurprisingly, we observed that predation had a higher irreplaceable mortality in tropical zones. In contrast, we observed low mortality from parasitoids and pathogens in both zones. Likewise, and very surprisingly, there were no mortality differences in natural enemy and non-natural enemy factors based on whether the insect species was native or non-native.

We observed that mortality from non-natural enemy factors was significantly greater than from natural enemies. Similar results were observed by Peterson et al. (2009), but here we used data from 268 life table studies instead of the 73 previously analyzed. We also used different criteria by including life tables that only had two reported mortality factors (n = 58) (e.g., parasitism and unknown). In contrast to our results, other studies of compiled life tables suggested that natural enemies were the most frequent mortality source for herbivorous insect species (Cornell & Hawkins, 1995; Hawkins, Cornell & Hochberg, 1997). However, these other studies presented few data from tropical zones and used a different analytical technique.

Abiotic factors such as temperature and rainfall are well-known causes of insect mortality (Birch, 1957; Krebs, 2014; Varella et al., 2015; Santos et al., 2020). However, because the MDLT analysis had actual measurements of mortality by cause, our results suggest that both temperate and tropical insects experience large mortality from abiotic factors, when considering both mortality in the presence of other factors and irreplaceable mortality. Abiotic factors are well known for their important role in plant-herbivorous insect interactions and how they affect diversity patterns in climate zones (Du et al., 2020).

Abiotic factors can also influence natural-enemy survival and establishment in different environments (Beirne, 1970; Norris, Memmott & Lovell, 2002). Furthermore, plant defenses and plant nutrition in general (which we grouped in the non-natural enemy mortality category) can negatively affect herbivore development and the performance of their natural enemies (Karban & Myers, 1989; Price et al., 1990; Coley & Barone, 1996; Havill & Raffa, 2000; Dyer & Coley, 2002).

We observed that irreplaceable mortality from predators in tropical zones was 3.7-fold greater than in temperate zones. Novotny et al. (2006) concluded that predation pressure in tropical conditions was 18-fold greater than in temperate conditions (mostly from predation by ants). Moreover, they suggested that plant diversity is crucial for large predator populations in tropical zones. This is supported in our analysis in Cluster 1, which is characterized by high irreplaceable mortality from predators (Fig. 4). Other studies also reiterate the importance of plant-community diversity for predator populations in tropical zones (Coley & Aide, 1991; Coley & Barone, 1996; Dyer & Coley, 2002). In addition to higher plant diversity, the greater stability of annual biomass production and environmental conditions in the tropics can be critical to sustaining both abundance and diversity of predator species (Paine, 1966; Hawkins, Cornell & Hochberg, 1997; Langellotto & Denno, 2004; Archibald et al., 2010; Jacquot et al., 2019).

Somewhat surprisingly, we observed that irreplaceable mortality for parasitoids was not different between temperate and tropical zones. Some researchers have argued that egg parasitoids are more diverse and abundant in the tropics than in temperate zones (Rabinowitz & Price, 1976; Morrison, Auerbach & McCoy, 1979; Hawkins, Cornell & Hochberg, 1997), while others argue that parasitoids are more important in temperate zones (Hawkins, Cornell & Hochberg, 1997). Similarities in parasitoid occurrence between climate zones were observed by Hawkins (1994) for more than 1,200 hosts worldwide and by Stireman et al. (2005) who analyzed parasitism data from 15 Lepidoptera rearing programs between southern Canada and central Brazil. The superfamily Ichneumonoidea tends to have lower population densities in tropical compared to temperate zones because of factors such as plant defenses and competition, which make hosts less available to these parasitoids in the tropics (Timms, Schwarzfeld & Sääksjärvi, 2016).

Although parasitoids can be important agents of insect mortality, we observed low values in both temperate and tropical zones. Mortality from parasitism would most likely have been largely replaced by other mortality causes if parasitism was not present. This can be seen in Cluster 3, in which high irreplaceable mortality occurred from non-natural enemy factors and low mortality from natural enemy factors (Fig. 4). We recognize that estimating mortality from natural enemies can be challenging because it depends on the methods from each study (e.g., determining parasitism or disease mortality by dissecting sample hosts, collecting hosts in situ and then rearing them in a laboratory, or detecting signals in the host body) (Van Driesche et al., 1991; Royama, 2001). Furthermore, host voltinism can also influence the effect of parasitism on host populations (Lawrence, 1990; Zaman et al., 2010). Our analysis had 46 univoltine/bivoltine and 39 multivoltine host species. It is important to note that for host species with numerous generations per season or year, low mortality rates from parasitism have the potential eventually to result in large population reductions and thus can flatten population growth curves (Knipling, 1992).

A potentially important finding from our study is that there were no statistical differences in natural enemy and non-natural enemy mortality between non-native and native insects. The fact that non-native insects do not experience lower mortality from natural enemies compared to native insects has been noted (Hawkins, Cornell & Hochberg, 1997). Colautti et al. (2004) discuss impacts of non-native invasive species and why their increased abundance in new habitats is not exclusively related to lower mortality from natural enemies (i.e., the Natural Enemy Release Hypothesis). The Natural Enemy Release Hypothesis has been supported by a number of cases showing that many introduced species grow faster or survive longer in their invaded vs their native localities (Torchin & Lafferty, 2009; Prior et al., 2015). However, there is contrary evidence (Agrawal & Kotanen, 2003; Colautti et al., 2004; Paula et al., 2020), and some researchers have argued that outcomes are highly variable and are better supported by other explanations such as climatic variables, human disturbance, or study methodology (Keane & Crawley, 2002; Jeschke et al., 2012; Meijer et al., 2016). Our results do not support the Natural Enemy Release Hypothesis, but we will present a more extensive analysis of our results as they relate to that hypothesis in another paper.

We observed low insect mortality by pathogens. This was not surprising considering that the impact of pathogens is limited by abiotic factors (e.g., temperature, UV radiation, humidity). Although tropical environments have high densities of arthropod pathogens (Mahé et al., 2017), we did not observe a difference between the climate zones.

Conclusions

Understanding how mortality factors affect insect population dynamics is challenging because these factors are variable, regardless of location (Barbosa, Letourneau & Agrawal, 2012). However, the MDLT and associated analytical techniques are robust biodemographic tools (Carey & Roach, 2020) to evaluate mortality dynamics on insect populations (Pereira et al., 2007; Asiimwe et al., 2007; Buteler et al., 2015; Varella et al., 2015; Achhami et al., 2020). In light of this, our study provides valuable information and suggests that in tropical conditions evaluating and using predators as pest management tactics should be a high priority. Furthermore, because parasitoid mortality was low in both tropical and temperate zones, we need to better understand the impact of parasitoid mortality, especially as it relates to biological control programs and integrated pest management.

Limitations and insights

Our goal in examining published life tables of insects is ultimately to address the crucially important topic of how insects die and contribute to the long discussion and debate regarding the relative importance of biotic and abiotic factors in population dynamics. Any metadata analysis is limited by the availability of data, which in turn reflects potential priorities or biases of funding agencies and researchers. For example, in this analysis of insect mortality, data are not reflective of insect diversity by taxon, numbers, or biomass. Similarly, the data themselves are subject to the limitations or biases associated with methods and techniques used in their acquisition. For instance, none of the life table studies we examined included a category for mortality from competition, yet we are confident that mortality through competition does occur (especially with non-native species). To illustrate, in our (LGH) experimental work on competition from Aedes albopictus and Aedes triseriatus (Novak et al., 1993) and competition from the blow flies Phormia regina, Lucilia sericata, and Chrysomya rufifacies (MacInnis & Higley, 2020), competitive superiority of introduced species vs native species was documented. Data limitations do not necessarily invalidate the meta-analysis any more than the individual studies might be invalidated, but recognizing these limits requires that we explicitly consider limitations in our interpretation of meta-analysis results.

Given the expense and experimental difficulties associated with life table studies, we would expect these to be directed towards species of economic importance (largely agricultural pests) and species in which causes of death are estimable. These predictions are accurate given that 55.2% of the studies are in crops and more than half of the studies focused on Lepidoptera (54.21%) and almost a quarter on Coleoptera (21.50%), almost exclusively of economic species in crop and forestry systems (Table S2). Our a priori expectations that abiotic mortality would be most important in temperate regions and biotic mortality in tropical regions were not accurate.

Some broader conclusions are possible within the limits of available data. Focusing just on Lepidoptera of economic importance, we note that like many agricultural insect pests (Hill, 1978) these life tables include a combination of native and introduced species, variable host associations but a general bias towards host specificity, and herbivores. Regarding habitat, in temperate and tropical regions agricultural fields are highly simplified ecosystems, most characteristic of an early successional community. In this context, our finding that irreplaceable mortality from abiotic factors was greater than biotic mortality seems less surprising. Indeed, the important comparison might be life table data from lepidopterans in agricultural settings vs lepidopterans in complex communities, like a prairie or rainforest ecosystem, to see if there is a shift to greater proportional mortality from natural enemies.

Despite this, we believe that because of the agroecosystem bias of the life table studies with economically important species, we can come to some intriguing conclusions. First, mortality from parasitoids does not seem to be as effective as contemporary narratives, efforts, and funding suggest. Second, mortality from predation seems to be very important in tropical agricultural systems. Third, mortality, especially irreplaceable mortality, from abiotic factors seems to be very important in both zones, which has direct implications for the efficacy of biological control and integrated pest management programs. Indeed, if irreplaceable mortality from abiotic factors is generally high in agroecosystems and irreplaceable mortality from natural enemies is generally low, it directly challenges current emphases to manage insect pests using biological control (Peterson et al., 2009).

However, we also believe the limitations mentioned above preclude generalizations at this time for insects in natural ecosystems. Consequently, there is a clear need for much more study of insect mortality in these complex systems. This is especially true given that habitat loss, which, sensu lato, is arguably the most devastating insect mortality factor and typically falls outside the life table arena. At a time of climate change, Anthropocene extinction events, and potential for substantial reductions of insect diversity and abundance, the need for objective documentation, including life table analysis, of insect mortality in natural and managed ecosystems has never been greater.

Supplemental Information

Supplemental Information 1 Publications with life tables used in the study.

Click here for additional data file.

Supplemental Information 2 Life tables analyzed.

Click here for additional data file.

Additional Information and Declarations

Competing Interests

Author Contributions

Data Availability

The authors declare that they have no competing interests.

José R. L. Pinto conceived and designed the experiments, performed the experiments, analyzed the data, prepared figures and/or tables, authored or reviewed drafts of the paper, and approved the final draft.

Odair A. Fernandes analyzed the data, authored or reviewed drafts of the paper, and approved the final draft.

Leon G. Higley analyzed the data, authored or reviewed drafts of the paper, and approved the final draft.

Robert K. D. Peterson conceived and designed the experiments, performed the experiments, analyzed the data, prepared figures and/or tables, authored or reviewed drafts of the paper, and approved the final draft.

The following information was supplied regarding data availability:

The data are in the Supplemental Files.

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
