# Peer review of "Do patterns of insect mortality in temperate and tropical zones have broader implications for insect ecology and pest management?"

_PeerJ, doi:10.7717/peerj.13340_

## Round 0.1 · original submission · Major Revisions

· Academic Editor

Major Revisions

This is an interesting paper, but requires some more substantive thought. In particular, using 'tropical' versus 'temperate' dichotomy. I agree with the first reviewer here, that the cutoff is extremely arbitrary - why not do a latitudinal comparison? A more nuanced assessment of a latitudinal response is required here.

Reviewer 1 ·

Basic reporting

The authors provide and interesting and insightful study of insect mortality patterns at broad geographic scale. However, aspects of this manuscript are difficult to understand, and there are some weaknesses in design that need to be addressed.
To classify the studies in their data set, the authors use 39 degrees latitude as a dividing line between the tropics and temperate zone. They state that this is derived from the Koppen-Geiger climate classification scheme. However, the K-G scheme relies only on temperature, seasonality, and amount of precipitation. It does not utilize latitude at all. Why the authors chose 39 degrees latitude requires clarification.
Designating a dividing line between the tropics and temperate zone by latitude alone will necessarily be a bit fraught and arbitrary. However, 39 degrees is very far north to draw this line. Traditionally, viewed only by latitude, the tropics begin at 23 degrees and the subtropics begin at 35 degrees. In North America, where the majority of the studies in this data set occurred, any site (excepting coastal climates) north of 35 degrees experiences regular freezes in the winter. Closer to 39 degrees, winters are severe. In a study of insect mortality factors, the annual freeze line is the most biologically reasonable place to divide temperate from tropics as it should partially determine both abiotic mortality and the influence of the third trophic level on insect mortality. I recommend either redefining your boundaries and re-analyzing the data accordingly or redefining your terminology.

Ln 34: replace “for” with “from”
Ln 77: replace “for” with “than”
Ln 90: Table S1 provides 51 not 56 studies from “tropical” locales. In the Excel spreadsheet, 67 (not 56) of the 107 underlying studies lie south of 39 degrees latitude.
Ln 176 – 178: Is this a test against all sources of natural enemy mortality combined? Please provide the value for mean irreplaceable mortality of all natural-enemy factors combined.
Ln 185 – 188: This statement is extremely confusing as written. I cannot parse what you are trying to convey. Please re-write and have an outside of your lab reader evaluate for clarity.
Ln 203 – 209: This belongs in the Methods.
Ln 224 – 226: I would be more convinced if you stress tested this result by seeing if it also applies to truly tropical sites: ie. those south of 23 degrees latitude.
Limitations and Insights: This is a well written and insightful section.
Figure legends are insufficient and unclear and in the case of Figure 4, fragmentary.

Experimental design

Designating a dividing line between the tropics and temperate zone by latitude alone will necessarily be a bit fraught and arbitrary. However, 39 degrees is very far north to draw this line. Traditionally, viewed only by latitude, the tropics begin at 23 degrees and the subtropics begin at 35 degrees. In North America, where the majority of the studies in this data set occurred, any site (excepting coastal climates) north of 35 degrees experiences regular freezes in the winter. Closer to 39 degrees, winters are severe. In a study of insect mortality factors, the annual freeze line is the most biologically reasonable place to divide temperate from tropics as it should partially determine both abiotic mortality and the influence of the third trophic level on insect mortality. I recommend either redefining your boundaries and re-analyzing the data accordingly or redefining your terminology.

Validity of the findings

See above

·

Basic reporting

This manuscript meets all requirements for basic reporting. It is is clearly written and it follows PeerJ guidelines.

Experimental design

The methods used for this review are original, first developed by the same authors group and to me they seem to be adequate and well described.
My only recommendation will be to briefly state which were the criteria to select the papers used in the review. I understand that they are in the references, but it would be easier for the reader if they were listed in the text.

Validity of the findings

The results are very interesting and address issues that have not been tested in ecological theory. The approach seems original to me and the literature support for the analysis is outstanding.

Additional comments

This manuscript represents an impressive review of the literature on life table studies (107 publications, 268 life tables). Causes of death were analyzed and irreplaceable mortality by cause was estimated using multiple decrement life tables. Comparisons are made between temperate and tropical zones, as well as between native and non-native species. The research work is a logical continuation of the work by Peterson et al (2009), where applying the same methods expands the database to make this type of comparisons. In my opinion this is an excellent paper. The Multiple Decrement Life Table analysis provided as supplementary material is extremely valuable.
The results challenge general ecological expectations, such as the greater importance of abiotic factors in temperate climates, or the greater susceptibility of native insects, compared to non-native ones, to natural enemies. However, these general expectations have not always been empirically validated. For me this highlights the importance of this study and for that reason I recommend accepting it for publication.

Specific comments.
L 82. “… so we can compare” instead of “… “so we are able to compare”.
L 104-105. Could you state which were those basic criteria? I understand that they are in the references, but it would be easier for the reader if they were listed here. For example, I assume you only consider open field life-table studies and do not consider laboratory experimental studies.
L 177-178. Include the mean percent (±SE) of irreplaceable mortality for the "all types of natural-enemy mortality" factor. According to figure 2 is aprox. 21%
L 187-188. Delete “which were all natural enemies, all parasitoids, pathogens, and all non-natural enemy factors”. There is no need to repeat.
L 210-220. I am not sure about the implications of this cluster analysis. Does it really have a biological meaning? Honduras is a tropical country and is in cluster 2. Is hard to explain why in Canada and Kenya, pathogens have a more relevant role than in other locations. Perhaps you might reconsider to delete this non-hierarchical multivariate k-means clusters analysis.
L 224 – 226. Could this be explained because this category includes "other causes" or not identified causes? For example, the cause of death might be a pathogen, but without symptoms that allow its identification. Or the cause of death might be physiological, intoxication or other cause that in insect life -table studies are not considered.
L 316-316. This recommendation should be taken with caution, because in tropical zones predators generally are already present. The introduction of predators can cause great disturbances. Perhaps conservation measures.
L 317-319. The evidence on the effect of parasitoids producing high levels of parasitism (= mortality) is extensive, although not in life table studies. Do you think this could be an explanation for the low mortality caused by parasitoids?
Legend of Figure 4 was cut or is incomplete

---

## Round 0.2 · Minor Revisions

· Academic Editor

Minor Revisions

I agree with the reviewer's assessments of your edits. There are still some issues to consider in your final revision.

Whilst I still do think that the tropical - temperate comparisons is quite broad, and would prefer to see a more fine-scale approach - I acknowledge that this is more based on the many ways to 'peel the potato', and the approach you take is valid and will be a substantive contribution to the literature.

Reviewer 1 ·

Basic reporting

Ok.

Experimental design

The authors have addressed my previous criticisms. Although I still think more consideration could be given to dissecting the temperate – tropical “dichotomy”, and I think the clustering based on political boundaries is problematic (see below). On balance I think this is a worthy study.

The K-means clustering of countries based on the studies conducted therein is a bit troubled. The authors include all countries in the study, however many of these countries are only represented by 1 to 3 lifetable studies. It is stretch to state that these few studies present a framework from which to generalize at the country level. In fact, most of the countries in these clusters are underpinned by 5 or fewer lifetables. Probably these countries should be dropped from the analysis. (And the 2 studies from Hawaii, should probably not be lumped with studies from the “United States”.) Alternatively, a more logical approach in my view, would be to group studies by biogeography rather than political boundaries.

93 – 96: In fact, you did include a sub-tropical group, you just did not split it out. Studies from the subtropics constitute a little over 1/3 of your tropical group. This raises the question of how reflective your results are of tropical systems. It was for this reason that I previously recommended stress testing your results by conducting an analysis restricted to those studies that took place in unquestionably tropical versus clearly temperate environments. If the patterns hold, even if you lack sufficient power for significance, the reader can have more confidence in your findings. This approach would be particularly important for results that are surprising. For example, the lack of difference in irreplaceable mortality for parasitoids or the overall balance of abiotic and biotic mortality. Do these values shift in the direction predicted by existing theory when a more tropical set of life tables is examined?

Validity of the findings

The authors have addressed my previous criticisms. Although I still think more consideration could be given to dissecting the temperate – tropical “dichotomy”, and I think the clustering based on political boundaries is problematic (see below). On balance I think this is a worthy study.

The K-means clustering of countries based on the studies conducted therein is a bit troubled. The authors include all countries in the study, however many of these countries are only represented by 1 to 3 lifetable studies. It is stretch to state that these few studies present a framework from which to generalize at the country level. In fact, most of the countries in these clusters are underpinned by 5 or fewer lifetables. Probably these countries should be dropped from the analysis. (And the 2 studies from Hawaii, should probably not be lumped with studies from the “United States”.) Alternatively, a more logical approach in my view, would be to group studies by biogeography rather than political boundaries.

93 – 96: In fact, you did include a sub-tropical group, you just did not split it out. Studies from the subtropics constitute a little over 1/3 of your tropical group. This raises the question of how reflective your results are of tropical systems. It was for this reason that I previously recommended stress testing your results by conducting an analysis restricted to those studies that took place in unquestionably tropical versus clearly temperate environments. If the patterns hold, even if you lack sufficient power for significance, the reader can have more confidence in your findings. This approach would be particularly important for results that are surprising. For example, the lack of difference in irreplaceable mortality for parasitoids or the overall balance of abiotic and biotic mortality. Do these values shift in the direction predicted by existing theory when a more tropical set of life tables is examined?

---

## Round 0.3 · accepted · Accept

· Academic Editor

Accept

I am happy with your responses to the extensive reviews provided. I believe that the manuscript is now more robust and will make a substantive contribution to a number of research fields.

Congratulations on getting this research published.